# Learning Invariant Representations with a Nonparametric Nadaraya-Watson Head

**Alan Q. Wang**
Cornell University

**Minh Nguyen**
Cornell University

**Mert R. Sabuncu**
Cornell University

## Abstract

Machine learning models will often fail when deployed in an environment with a data distribution that is different than the training distribution. When multiple environments are available during training, many methods exist that learn representations which are invariant across the different distributions, with the hope that these representations will be transportable to unseen domains. In this work, we present a nonparametric strategy for learning invariant representations based on the recently-proposed Nadaraya-Watson (NW) head. The NW head makes a prediction by comparing the learned representations of the query to the elements of a support set that consists of labeled data. We demonstrate that by manipulating the support set, one can encode different causal assumptions. In particular, restricting the support set to a single environment encourages the model to learn invariant features that do not depend on the environment. We present a causally-motivated setup for our modeling and training strategy and validate on three challenging real-world domain generalization tasks in computer vision.

## 1 Introduction

Machine learning models often fail when there is significant distribution shift. The goal of domain generalization is to be able to perform well with new distributions [21, 56, 71]. In this work, we are interested in settings where multiple domains/environments are available during training and we have access to environment indicators. A popular way to tackle domain generalization in this setting is to learn representations that are invariant across environments [20, 41, 60]. The hope is that such representations will work well in, or are transportable to, unseen environments. This invariance is often encoded via constraints on a learned predictor which aligns its behavior across environments; often, these conditions are derived using causal reasoning and/or by making assumptions about the data-generating process [39].

In a parametric setting, almost all existing methods enforce these constraints by training a single model and adding a regularizer on top of a standard predictive loss [1, 5, 18, 20, 27, 48, 54, 58]. Most notably, invariant risk minimization (IRM) enforces the representations to be such that the optimal classifier on top of those representations is the same across all environments. Other examples include enforcing the layer activations of the predictor to be aligned across environments [48], enforcing the predictor to be calibrated across environments [54], and enforcing gradients of the predictor to be aligned across environments [46]. Often, optimizing these constraints demand approximations or relaxations that undermine the efficacy of the approach [22].

In this work, we take a different approach using a nonparametric strategy based on a recently-proposed Nadaraya-Watson (NW) head [55]. Instead of computing the class probability directly from an input query, the NW head makes a prediction by comparing the learned representations of the query to the elements of a support set that consists of labeled data. Thus, the NW prediction is computed *relative to other real datapoints* in the support set, with the support set providing a degree of flexibility not

possible with parametric models. In particular, one can manipulate it during training in a way which restricts the types of comparisons that the model can make.

In this work, we manipulate the support set during training to encode causal assumptions for the purposes of learning invariant representations. Specifically, restricting the support set to be drawn from a single environment precludes the possibility of using environment-specific features to make a prediction for a given query. We show that this setup is causally-motivated and relates to existing causal frameworks. Furthermore, we show that this training strategy leads to competitive to superior results compared to state-of-the-art parametric baselines.

Our contributions are as follows:

- We present causally-motivated assumptions for domain generalization which justify our modeling and training strategy.
- We present a novel approach to invariant representation learning using the nonparametric Nadaraya-Watson head, which can account for causal assumptions by manipulating a support set. In particular, we propose a training strategy which, unlike competing baselines, has *no invariance hyperparameter to tune*.
- We validate our approach on several datasets and demonstrate competitive results compared to state-of-the-art parametric baselines.

## 2 Related Works

### 2.1 Domain Generalization and Invariant Representations

Domain generalization seeks to make models robust to unseen environments and is an active area of research [21, 56, 71]. One line of work augments or synthetically-generates additional training images to increase robustness of learned features to unseen environments [59, 63–65, 70]. In particular, LISA uses a mixup-style [67] interpolation technique to generate augmented images, which the authors demonstrate improves out-of-distribution robustness [64]. Another line of work broadly seeks to align features across distributions. Deep CORAL aligns correlations of layer activations in deep neural networks [48], and other works minimize the divergence of feature distributions with different distance metrics such as maximum mean discrepancy [51, 32], an adversarial loss [14, 30], and Wasserstein distance [69]. Still other works approach the problem from the perspective of the gradients and optimization [12, 29, 34, 46, 57]. For example, Fish aligns the gradients from different domains [46].

One can also achieve domain generalization via learning invariant representations, which often requires reasoning about the data-generating process from a causal perspective to arrive at appropriate constraints [39]. Invariant causal prediction (ICP) formulates the problem from a feature selection perspective, where the goal is to select the features which are direct causal parents of the label [40]. Invariant Risk Minimization (IRM) can be viewed as an extension of ICP designed for deep, nonlinear neural networks. The IRM objective can be summarized as finding the representation $\varphi$ such that the optimal linear classifier's parameters $w^*$ on top of this representation is the same across all environments [1]. This bi-level program is highly non-convex and difficult to solve. To find an approximate solution, the authors consider a Lagrangian form, whereby the sub-optimality with respect to the constraint is expressed as the squared norm of the gradients of each of the inner optimization problems. Follow-up works analyzing IRM have raised theoretical issues with this objective and presented some practical concerns [16, 22, 42]. Various flavors of IRM have also been proposed by introducing different regularization terms [27, 54, 58].

### 2.2 Nonparametric Deep Learning

Nonparametric models in deep learning have received much attention in previous work. Deep Gaussian Processes [10], Deep Kernel Learning [62], and Neural Processes [24] build upon Gaussian Processes and extend them to representation learning. Other works have generalized $k$-nearest neighbors [37, 49], decision trees [68], density estimation [13], and more general kernel-based methods [15, 36, 66] to deep networks and have explored the interpretability that these frameworks provide. Closely-related but orthogonal to nonparametric models are attention-based models, most notably self-attention mechanisms popularized in Transformer-based architectures in natural language

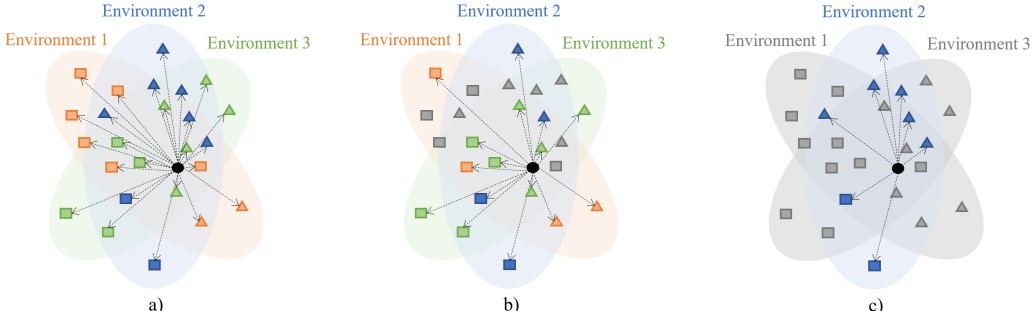

Figure 1: Illustration of proposed approach. Support set of labeled datapoints (square/triangle) from 3 environments lie in 3 regions in the feature space. Black circle denotes query datapoint with unknown label. a) The NW head models $P(Y|X)$ by making predictions as a function of distances to labeled datapoints in the feature space (visualized as dotted arrows) . b) Balancing comparisons across labels for all environments models $P^B(Y|X)$. c) Conditioning on a single environment models $P_e(Y|X)$.

processing [53] and, more recently, computer vision [11, 19, 38]. Nonparametric transformers apply attention in a nonparametric setting [24].

Recently, Wang et al. proposed the NW head [55], an extension of the classical NW model [2, 35, 61] to deep learning. In the NW head, the prediction is a weighted average of labels from a support set. The weights are computed from distances between the query and support features. The NW head can yield better calibration and interpretability, with similar accuracy compared to the dominant approach of using a parametric classifier with fully-connected layers. In this work, we leverage the NW head to encode causal assumptions via the support set. The interpretability and explainability benefits of the NW head carry over in this work; while not of primary focus, we explore these properties in the Appendix.

# 3    Preliminaries

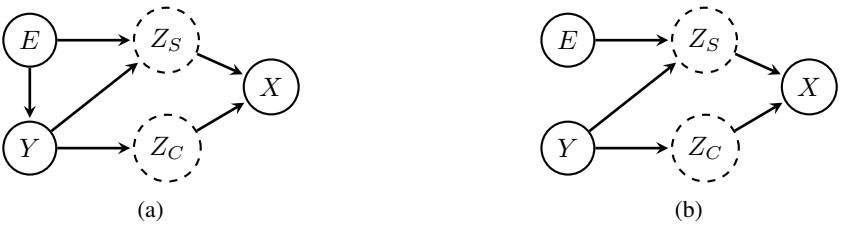

Figure 2: a) Causal Directed Acyclic Graph (DAG) we consider in this work. Solid nodes are observed and dashed nodes are unobserved. We assume an anti-causal setting where label $Y$ causes $X$, and $X$ has 2 causal parents: "style" features, $Z_S$, which are influenced by the environment $E$; and environment-independent "content" features of $X$, $Z_C$, which are causally influenced by the label $Y$. $E$ potentially influences $Y$. Both $E$ and $Y$ have direct influence on style features $Z_S$. b) Same DAG as a) with an intervention on $Y$. We note that $Y \perp\!\!\!\perp E \mid Z_C$ and $Y \not\perp\!\!\!\perp E \mid Z_S$.

**Problem Setting.** Let $X, Y$ denote a datapoint and its corresponding discrete class, and $E$ denote the environment (or domain) where $X, Y$ originates.[1] That is, the elements of the training dataset $\mathcal{D}_{tr} = \{x_i, y_i, e_i\}_{i=1}^N$ are drawn first by sampling the discrete random variable $e_i \sim P(E)$, and then sampling $x_i, y_i \sim P(X, Y \mid E = e_i) := P_{e_i}(X, Y)$. Our goal is to learn classifiers that will generalize to new, unseen environments.

**Assumptions.** We assume there exists a pair of latent causal parents of $X$: an environment-independent ("content") factor $Z_C$ and an environment-dependent ("style") factor $Z_S$.[2] We as-

---

[1]We assume the support of $E$, $\mathrm{supp}(E)$, is finite.

[2]This is sometimes referred to as style-content or causal decomposition [21, 42, 60].

sume the causal mechanism that generates $X$ from $(Z_C, Z_S)$ is injective, so that, in principle, it is possible to recover the latent features from the observations; i.e. there exists a function $g$ such that $g(X) = (Z_C, Z_S)$. We further assume that $g$ can be disentangled into $g_C$ and $g_S$, such that $(Z_C, Z_S) = g(X) = (g_C(X), g_S(X))$. The causal graph is depicted in Fig. 2a. Finally, we assume that if any $X = x$ has a non-zero probability in one environment, it has a non-zero probability in all environments.

**Motivation.** The motivating application in this work is image classification, where each realization of $E$ might represent a different site, imaging device, or geographical region where $X, Y$ are collected. For example, in medical imaging, different hospitals ($E$) may collect images ($X$) attempting to capture the presence of some disease ($Y$), but may differ in their imaging protocols which lead to differences in the image style features $Z_S$ (e.g. staining, markings, orientation). In addition, we allow $Z_S$ to be influenced by the label itself (for example, positive examples are more likely to be marked by a doctor or have specific staining than negative examples). Finally, the prevalence of $Y$ may be influenced by $E$ (for example, the prevalence of a disease may be higher in a certain hospital).

The goal is to find an estimator for $Y$ which relies only on the direct causal links between $Y$ and $X$ and not on any spurious associations between $E$ and $X$, as these may change in a new, unseen environment. That is, we seek an estimator which relies only on $Z_C$ and which is independent of $E$ or $Z_S$.

First, we note the direct causal dependence $E \to Y$. For example, a model can exploit this association by learning to over-predict majority classes in a certain environment. One way to remove the direct dependence is by intervening on $Y$, thus removing incoming edges to $Y$. This essentially corresponds to matching the environment-specific prior on $Y$ between environments, and results in the intervened graph in Fig. 2b.[3] Let us refer to any distribution which follows the DAG in Fig. 2b as $P_e^B(X, Y)$.

Second, we observe that there is a potential non-causal association flow between $E$ and $Y$ through the colliders $X$ and $Z_S$, when either one of these are conditioned on (i.e. are observed). An estimator which relies on $Z_S$ potentially leaks information from $E$, and this is unstable in new environments. Reading d-separation on this intervened graph, we infer that $Y \perp\!\!\!\perp E \mid Z_C = g_C(X)$ and $Y \not\perp\!\!\!\perp E \mid Z_S = g_S(X)$, that is:

$$P_e^B(Y \mid g_C(X)) = P_{e'}^B(Y \mid g_C(X)) \ \ \forall e, e' \in E. \tag{1}$$

In words, this assumption states that the probability of $Y$ given the environment-invariant parts of $X$ is the same across any environment $e \in E$.

Thus, we seek an estimator that 1) enables interventions on $Y$ such that the direct dependence $E \to Y$ can be removed, and 2) can further encode the assumption in Eq. (1).

## 4  NW Head for Invariant Prediction

Given a datapoint $x$, support set $\mathcal{S} = \{x_i, y_i\}_{i=1}^{N_s}$, and parameters $\phi$, the NW head estimates $P(Y = y \mid X = x)$ by outputting a prediction formulated as a weighted sum of support set labels, where the weights are some notion of similarity in the feature space [55]:

$$\hat{P}(Y = y \mid X = x; \phi) := f_\phi(x, \mathcal{S}) = \frac{\sum_{i=1}^{N_s} \exp\left\{s(\phi(x), \phi(x_i))\right\} \vec{y_i}}{\sum_{j=1}^{N_s} \exp\left\{s(\phi(x), \phi(x_j))\right\}}. \tag{2}$$

Here, $\vec{y}$ is the one-hot encoded version of $y$ and $s(\cdot, \cdot)$ is a similarity/kernel function that captures the similarity between pairs of features. In this work, we set $s$ as the negative Euclidean distance. A graphical depiction is shown in Fig. 3.

Manipulating the support set can encode certain causal assumptions. We consider two types of manipulations:

1. Balancing classes in $\mathcal{S}$, denoted $\mathcal{S}^B$ (see Fig. 1b). This can be interpreted as an intervention on $Y$, and removes the dependence on $E \to Y$, i.e.:

$$\hat{P}^B(Y = y \mid X = x; \phi) := f_\phi(x, \mathcal{S}^B). \tag{3}$$

---

[3]This may be interpreted as making the model robust to label shift, see [31, 44].

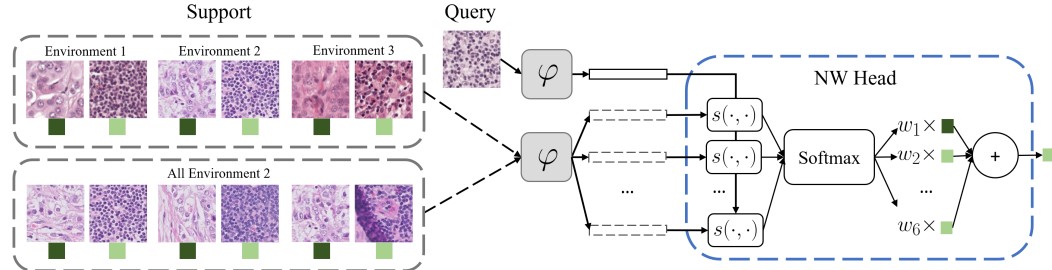

Figure 3: A depiction of the NW head on a tumor detection task. The NW head computes Euclidean distances $s(\cdot, \cdot)$ between query and support features, and uses the distances to weight the support labels. Colored squares represent labels. Diagram displays two different support sets. Top is unconditional support, where support data is drawn from the training data without knowledge of environment information. Bottom is an example of a manipulated support where all support data is drawn from a fixed environment (note similarity in color). Such a support set precludes the possibility of using environment-specific features to make a prediction.

2. Conditioning $\mathcal{S}$ on a single environment, denoted $\mathcal{S}_e$ (see Fig. 1c). This can be interpreted as conditioning the probability estimate on $E = e$, i.e.:

$$\hat{P}_e(Y = y \mid X = x; \phi) := f_\phi(x, \mathcal{S}_e). \tag{4}$$

Note that both balancing and conditioning can be achieved simultaneously, which we denote $\mathcal{S}_e^B$.

### 4.1 Objective and Optimization Details

Given a dataset of samples $\mathcal{D}_{tr} = \{x_i, y_i, e_i\}_{i=1}^N$, we wish to leverage the NW head as a conditional estimator for $Y$ conditioned on $Z_C = g_C(X)$, where $g_C(X)$ is characterized by Eq. (1). This necessitates an optimization over both $\phi$ and the space of functions $g_C$. Thus, we solve the following constrained maximum likelihood over $\phi$ and $g_C$:

$$\underset{\phi, g_C}{\operatorname{argmax}} \sum_{i=1}^N \log \hat{P}_{e_i}^B(y_i \mid g_C(x_i); \phi) \tag{5}$$

$$\text{s.t. } \hat{P}_e^B(y_i \mid g_C(x_i); \phi) = \hat{P}_{e'}^B(y_i \mid g_C(x_i); \phi), \quad \forall i \in \{1, ..., N\}, \ \forall e, e' \in E.$$

Note that Eq. (1) implies that $P_e^B(y_i \mid g_C(x_i)) = P^B(y_i \mid g_C(x_i))$. Thus, the objective is equivalent to unconstrained maximum likelihood under the assumption in Eq. (1).

Instead of solving for $g_C$ explicitly, we let both $\phi$ and $g_C$ be related by the composition $\varphi = \phi \circ g_C$, and set $\varphi$ to be the learnable mapping of the NW head, i.e. a neural network. Then, the objective becomes:

$$\underset{\varphi}{\operatorname{argmin}} \sum_{i=1}^N L(f_\varphi(x_i, \mathcal{S}_{e_i}^B), y_i) \tag{6}$$

$$\text{s.t. } f_\varphi(x_i, \mathcal{S}_e^B) = f_\varphi(x_i, \mathcal{S}_{e'}^B), \quad \forall i \in \{1, ..., N\}, \ \forall e, e' \in E,$$

where $L$ is the cross-entropy loss. To make the objective tractable, we consider two possible variants:

1. **Explicit.** Solve the optimization problem explicitly via a Lagrangian formulation:

$$\underset{\varphi}{\operatorname{argmin}} \sum_{i=1}^N L(f_\varphi(x_i, \mathcal{S}_{e_i}^B), y_i) + \lambda \sum_{e, e' \in E} \sum_{i=1}^N \|f_\varphi(x_i, \mathcal{S}_e^B) - f_\varphi(x_i, \mathcal{S}_{e'}^B)\|_2^2. \tag{7}$$

where $\lambda > 0$ is a hyperparameter.

2. **Implicit.** Relax the optimization problem into the following unconstrained problem:

$$\underset{\varphi}{\operatorname{argmin}} \sum_{e \in E} \sum_{i=1}^N L(f_\varphi(x_i, \mathcal{S}_e^B), y_i). \tag{8}$$

In this formulation, the constraint will be approximately satisfied in the sense that the model will be encouraged to predict the ground truth for a given image, which is identical across all environments. In practice, how well the solution satisfies the constraint will depend on model capacity, the data sample, and optimization procedure.

## 4.2 Optimization Details

During training, the support set $\mathcal{S}$ is drawn stochastically from the training set $\mathcal{D}_{tr}$, and all queries and support datapoints are passed through the feature extractor $\varphi$. For computational efficiency, instead of sampling a unique support mini-batch at the query-level, we sample a unique support at the mini-batch level. Thus, if $N_q$ and $N_s$ are the query and support mini-batch sizes respectively, the effective mini-batch size is $N_q + N_s$, instead of $N_q N_s$. For the implicit variant, we sample one support set for a given mini-batch of queries, forward pass through the NW head, and compute the loss in Eq. (8). For the explicit variant, we sample two support sets for a given mini-batch of queries, perform two independent forward passes through the NW head for each support set, and compute the loss in Eq. (7). As discussed in prior work [55], the support batch size is a hyperparameter analogous and orthogonal to the query batch size.

A technical point is that the set of labels in the support mini-batch must cover the set of labels in the query mini-batch. Thus, in our implementation, for $\mathcal{S}^B$, we cycle through all classes and randomly draw $N_c$ examples per class to include in the support. For tasks with a large number of classes, one can subsample from the total number of classes, so long as the sampled classes cover the set of query classes.

## 4.3 Inference modes

Similar to how the support set can be manipulated during training, we can also design different inference strategies corresponding to different configurations of the support set at test-time. We explore several different inference modes which are possible under the NW framework:

1. **Random.** Sample uniformly at random over the dataset, such that each class is represented $k$ times.
2. **Full.** Use the entire balanced training set.
3. **Ensemble.** Given the set of balanced features computed from Full mode, partition the computed features for all training datapoints by environment, compute the softmax predictions with respect to each environment balanced across labels, and average the predictions.
4. **Cluster.** Given the set of balanced features computed from Full mode, perform $k$-means clustering on the features of the training datapoints for each class. These $k$ cluster centroids are then used as the support features for each class. This can be viewed as a distillation of the full training set for efficient inference, with the caveat that the support set no longer corresponds to observed datapoints.

While Full, Ensemble, and Cluster require computing features for the entire support set, in practice these features and centroids can be precomputed. In our experiments, we find that Cluster mode can be a sufficient replacement to Full mode, while being computationally cheaper. These inference modes can be used interchangeably and flexibly. As an example, consider a workflow which would involve using Cluster mode to perform efficient inference, and then using Full mode on a select few (potentially problematic) test queries to understand model behavior.

## 4.4 Connections to Prior Work

Our assumptions in Eq. (1) are common across many works related to learning invariant predictors [40, 27, 54, 41, 26]. Representatively, under the binary classification setting, the IRM objective finds a representation function $\varphi$ which elicits an invariant predictor across environments $E$ such that for all $h$ that has a non-zero probability for $\varphi(X)$ in any (and all) environment(s):

$$\mathbb{E}_e[Y \mid \varphi(X) = h] = \mathbb{E}_{e'}[Y \mid \varphi(X) = h], \ \forall e, e' \in E.$$

Eq. (1) can be viewed as a generalization of this equality to multi-class settings.[4]

---

[4]This equality has also been called "sufficiency invariance" [60].

Table 1: Summary of Datasets.

| Dataset | # Classes | Env | # Envs | Architecture | Metric |
|---------|-----------|-----|--------|--------------|--------|
| Camelyon-17 | 2 | Hospital | 3 | DenseNet-121 | Average acc. |
| ISIC | 2 | Hospital | 3 | ResNet-50 | F1-score |
| FMoW | 62 | Region | 5 | DenseNet-121 | Worst-region acc. |

Table 2: Metric average $\pm$ standard deviation for all datasets (%). Higher is better. **Bold** is best and underline is second-best. Implicit / Explicit.

| Algorithm | Camelyon-17 | ISIC | FMoW |
|-----------|-------------|------|------|
| ERM [52] | $70.3_{\pm 6.4}$ | $58.2_{\pm 2.9}$ | $32.6_{\pm 1.6}$ |
| IRM [1] | $70.9_{\pm 6.8}$ | $57.9_{\pm 1.0}$ | $31.3_{\pm 1.2}$ |
| CORAL [48] | $72.4_{\pm 4.4}$ | $59.1_{\pm 2.2}$ | $31.7_{\pm 1.0}$ |
| Fish [46] | $74.7_{\pm 7e\text{-}2}$ | $64.4_{\pm 1.7}$ | $34.6_{\pm 0.0}$ |
| LISA [64] | $77.1_{\pm 6.5}$ | $64.8_{\pm 2.3}$ | $35.5_{\pm 1.8}$ |
| CLOvE [54] | $\underline{79.9}_{\pm 3.9}$ | $66.2_{\pm 2.2}$ | $\mathbf{40.1}_{\pm 0.6}$ |
| $\text{NW}^{\text{B}}$, Random | $71.7_{\pm 5.3}$ | $56.7_{\pm 1.4}$ | $31.1_{\pm 0.8}$ |
| $\text{NW}^{\text{B}}$, Full | $72.0_{\pm 6.7}$ | $61.9_{\pm 3.5}$ | $31.6_{\pm 0.9}$ |
| $\text{NW}^{\text{B}}$, Cluster | $70.6_{\pm 6.9}$ | $61.4_{\pm 2.3}$ | $31.3_{\pm 0.9}$ |
| $\text{NW}^{\text{B}}$, Ensemble | $71.9_{\pm 6.0}$ | $63.9_{\pm 3.8}$ | $32.2_{\pm 1.0}$ |
| $\text{NW}^{\text{B}}$, Probe | $69.2_{\pm 7.4}$ | $59.7_{\pm 2.5}$ | $29.9_{\pm 1.5}$ |
| $\text{NW}_{\text{e}}^{\text{B}}$, Random | $74.8_{\pm 8.4}$ / $75.3_{\pm 3.2}$ | $57.5_{\pm 1.9}$ / $55.0_{\pm 0.9}$ | $31.2_{\pm 0.7}$ / $30.9_{\pm 0.5}$ |
| $\text{NW}_{\text{e}}^{\text{B}}$, Full | $\mathbf{80.0}_{\pm 2.7}$ / $79.7_{\pm 1.9}$ | $69.6_{\pm 2.3}$ / $70.0_{\pm 1.0}$ | $35.0_{\pm 0.7}$ / $34.6_{\pm 0.4}$ |
| $\text{NW}_{\text{e}}^{\text{B}}$, Cluster | $78.6_{\pm 2.5}$ / $79.0_{\pm 1.4}$ | $\mathbf{71.1}_{\pm 1.7}$ / $\underline{71.0}_{\pm 1.0}$ | $33.9_{\pm 0.6}$ / $34.0_{\pm 0.3}$ |
| $\text{NW}_{\text{e}}^{\text{B}}$, Ensemble | $79.5_{\pm 2.6}$ / $79.6_{\pm 1.9}$ | $69.5_{\pm 2.2}$ / $69.8_{\pm 0.8}$ | $37.8_{\pm 0.9}$ / $\underline{38.2}_{\pm 0.4}$ |
| $\text{NW}_{\text{e}}^{\text{B}}$, Probe | $75.3_{\pm 7.3}$ / $75.8_{\pm 8.3}$ | $61.4_{\pm 3.1}$ / $63.4_{\pm 2.8}$ | $33.9_{\pm 1.5}$ / $32.7_{\pm 1.4}$ |

Furthermore, note that given the feature extractor $\varphi$, the NW mechanism $f$ is a nonlearnable classifier, whereas $w$ is learned in the IRM setting. Thus, our proposed objective can be interpreted as learning invariant features $\varphi$, where the *fixed classifier constraint is satisfied by construction*. This avoids the need to approximate the complex bilevel optimization problem with a regularizer which assumes convexity and requires computing the Hessian. Essentially, $f$ enforces invariance through the manipulation of the support set, providing a more intuitive and computationally simpler objective to optimize.

In the Experiments section, we compare IRM against a variant of our algorithm where we freeze the learned representations and finetune a linear classifier on top using the same training data. We find that our algorithm performs better than IRM on all datasets, suggesting that it captures invariant representations better than IRM.

## 5 Experiments and Results

### 5.1 Baselines

We compare against several popular and competitive baseline algorithms: empirical risk minimization (ERM) [52], invariant risk minimization (IRM) [1], deep CORAL [48], Fish [46], LISA [64], and CLOvE [54]. When available, results on baselines are pulled from their respective papers. Details on baseline algorithms are provided in the Appendix.

### 5.2 Datasets

We experiment on 3 real-world domain generalization tasks. Two are from the WILDS benchmark [23], and the third is a challenging melanoma detection task. Details on the datasets are summarized in Table 1, and further information is provided in the Appendix.

1. The Camelyon-17 dataset [4] comprises of microscopic images of stained tissue patches from different hospitals, where the label corresponds to presence of tumor tissue in patches and the environment is the hospital where the patch comes from.

2. The melanoma dataset is from the International Skin Imaging Collaboration (ISIC) archive[5]. The ISIC dataset comprises of dermoscopic images of skin lesions from different hospitals, where the label corresponds to whether or not the lesion is diagnosed as melanoma and the environment is the hospital where the image comes from. There is significant less positive examples and negative examples, with this label imbalance varying across environments (see Appendix).

3. The Functional Map of the World (FMoW) dataset [6] comprises of RGB satellite images, where the label is one of 62 building or land use categories, and the environment represents the year the image was taken and its geographical region.

## 5.3 Experimental Setup

For each model variant, we train 5 separate models with different random seeds, and perform model selection on an out-of-distribution (OOD) validation set. For WILDS datasets, we follow all hyperparameters, use model selection techniques, and report metrics as specified by the benchmark. This includes using a DenseNet-121 backbone initialized with pretrained ImageNet weights as $\varphi$ and no random augmentations for both datasets. Similarly for ISIC, we use a pretrained ResNet-50 backbone as $\varphi$ with no augmentations, and perform model selection on an OOD validation set. Due to significant label imbalance, we report F1-score instead of average accuracy.

For NW algorithms, we refer to models which balance classes (i.e. modeling Eq. (3)) as $NW^B$, and models which additionally condition on environment (i.e. modeling both Eq. (3) and Eq. (4)) as $NW_e^B$. For $NW_e^B$ models, we train explicit and implicit variants. For all NW algorithms, we perform evaluation on all inference modes. In addition, for completeness, we experiment on a variant where we freeze the feature extractor and finetune a linear probe on top of the learned representations on the same training data $\mathcal{D}_{tr}$, which we refer to as "Probe". As an example, the implicit variant of $NW_e^B$ is trained on Eq. (8), where the support set is balanced across classes (B) and conditioned on an environment (e).

We set $N_c = 8$ for Camelyon-17 and ISIC and $N_c = 1$ for FMoW. An analysis of this hyperparameter is provided in the Appendix. The query batch size $N_q$ is set to 8 for all NW experiments. For Random and Cluster inference modes, we set $k = 3$. This was chosen based on prior work [55], where $k = 3$ was shown to balance good error rate performance with computational efficiency. For explicit variants, we tune $\lambda$ for all datasets via grid search on a held-out validation set. Full hyperparameter details are provided in the Appendix.

All training and inference is done on an Nvidia A6000 GPU and all code is written in Pytorch.[6]

## 5.4 Results

Table 2 shows the main results. We find that on Camelyon-17 and ISIC datasets, $NW_e^B$ with Full mode outperforms all baselines and variants we consider. In addition, $NW_e^B$ variants typically have lower variance across random seeds as compared to baselines. For FMoW, $NW_e^B$ with Ensemble mode performs around 2% lower than the best performing baseline, CLoVE. We observe that the most computationally-efficient inference mode, Cluster, performs comparably to Full mode for $NW_e^B$ models, and is in fact the highest-performing model for ISIC. Thus, we conclude that the support set can be an efficient replacement for Full.

For ISIC, we find that almost all $NW^B$ modes (except Random) perform $\sim 3\%$ better than ERM. This may be attributed to balancing classes across environments, which we suspect has added benefit for highly imbalanced tasks. In contrast, this boost is less apparent for Camelyon-17, which has relatively balanced classes. As an ablation, we compare $NW^B$ against an NW variant without class-balancing in the Appendix. $NW_e^B$ further improves over $NW^B$ by $\sim 7\%$. Exploring further, we compare $NW_e^B$ against an ERM variant with balanced classes per environment, which we denote $ERM^B$. This

---
[5]https://www.isic-archive.com
[6]Our code is available at `https://github.com/alanqrwang/nwhead`.

achieves $63.0 \pm 2.5$, which is on-par with NW$^\text{B}$. This is expected as the theoretical assumptions are the same for both models.

Comparing implicit to explicit variants of NW$_\text{e}^\text{B}$, we do not find much difference in explicitly enforcing Eq. (1), although we do observe significantly lower variances across model runs. Generally, we find the slight performance gain of explicit training to not be worth the computational overhead of doubling the number of support set forward passes per gradient step and tuning the hyperparameter $\lambda$.

While not the highest-performing, we highlight the consistent 1-5% improvement of Probe models over IRM, indicating that NW$_\text{e}^\text{B}$ may be better at capturing invariant features. However, other non-parametric inference modes still outperform Probe, possibly indicating that the learned features are more suitable for NW-style classifiers.

## 6   Discussion and Limitations

There are several advantages of the proposed NW approach over previous works. First, the implicit training strategy in Eq. (8) has no hyperparameter to tune, while remaining competitive with and often outperforming state-of-the-art baselines which all require tuning a hyperparameter coefficient in the regularized loss. Second, the NW head enables interpretability by interrogating nearest neighbors in the feature space. Since these neighbors directly contribute to the model's prediction (Eq. (2)), interrogation enables a user to see what is driving the model's decision-making. This not only allows for greater model transparency, but also enables interrogating the quality of the invariant features. We explore this capability in Section H in the Appendix. Note this this degree of transparency is not present in parametric baselines. Lastly, from an intuitive standpoint, we believe our non-parametric approach to enforcing invariance across environments is more natural than baseline methods, since an environment is encoded by manipulating the support set to contain real samples only from that environment. Other baseline methods resort to proxy methods to enforce invariance [54, 48, 27].

One important limitation of our method is computational (see Appendix for analysis of runtimes). The proposed approach requires pairwise comparisons, which scales quadratically with sample size. Practically, this means passing a separate support mini-batch in addition to a query mini-batch at every training iteration. This limitation is compounded for explicit variants, in which two sets of support sets must be drawn independently. Future work may explore more computationally-efficient training strategies. At inference time, Full, Cluster, and Ensemble modes are expensive procedures which require computing features for the entire support set, although precomputing features can mitigate this. However, we argue that in high-risk, safety-critical domains like medical imaging, high inference throughput may not be as important as performance, interpretability, and robustness.

We expect the proposed approach to work well with tasks that have several (and diverse) sets of examples per label class in each environment. If this is not the case, as in the FMoW dataset, the resulting model will be sub-optimal. In particular, in the extreme case where no example is present for a specific class in a given environment, constructing a support set with labels that cover the ground truth label of the query images will not always be possible. This will, in turn, impact performance.

## 7   Conclusion

We presented a nonparametric strategy for invariant representation learning based on the Nadaraya-Watson (NW) head. In the NW head, the prediction is made by comparing the learned representations of the query to the elements of a support set that consists of labeled data. We demonstrated two possible ways of manipulating the support set, and demonstrate how this corresponds to encoding different assumptions from a causal perspective. We validated our approach on three challenging and real-world datasets.

We believe there are many interesting directions of further research. First, our treatment is restricted to classification tasks. Future work may explore an extension to the regression setting. Second, it can be interesting to explore adaptation to the test domain, given additional information. For example, reweighting the occurrence of samples per label could provide improved results given knowledge about the edge $E \rightarrow Y$ in the test distribution. One can further envision implementing the proposed method in settings where there are previously unseen test time labels/tasks. Finally, we are interested in replacing the fixed similarity function with a learnable kernel.

## Acknowledgements

Funding for this project was in part provided by the NIH grant R01AG053949, and the NSF CAREER 1748377 grant.

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

# A    Description of Baselines

- Empirical risk minimization (ERM) [52] minimizes the sum of errors across domains and examples.

- Invariant risk minimization (IRM) [1] learns a feature representation such that the optimal linear classifier on top of that representation matches across domains. For WILDS datasets, we pull baseline performance from [54]. For ISIC, we use the implementation from [16].

- Deep CORAL [48] penalizes differences in the means and covariances of the feature distributions (i.e., the distribution of last layer activations in a neural network) for each domain. For WILDS datasets, we pull baseline performance from [54]. For ISIC, we use the implementation from [16].

- Fish targets domain generalization by maximizing the inner product between gradients from different domains. For WILDS datasets, we pull baseline performance from the original paper. For ISIC, we use the implementation from [16].

- LISA augments the set of training data by randomly performing two types of mixup-style [67] interpolations: intra-label (same label, different domain) and inter-label (same domain, different label). For WILDS datasets, we pull baseline performance from the original paper, and train their implementation on the ISIC dataset.

- CLOvE [54] finds an invariant classifier by enforcing the classifier to be calibrated across all training domains. While the original paper proposes several model variants leveraging this idea, we report their best-performing variant, which starts with a trained CORAL model and finetunes the weights using a regularized cross-entropy loss. The regularizer aggregates Maximum Mean Calibration Error (MMCE) [28] over all training domains. For WILDS datasets, we pull baseline performance from [54]. As their implementation is not publicly-available, we implement it for ISIC.

# B    Description of Datasets

Representative examples of the 3 datasets are shown in Fig. 4.

- **Camelyon-17.** We use Camelyon-17 from the WILDS benchmark [4, 23], which provides 450,000 lymph-node scans sampled from 5 hospitals. Camelyon-17 is a medical image classification task where the input $x$ is a $96 \times 96$ image and the label $y$ is whether there exists tumor tissue in the image. The environment denotes the hospital that the patch was taken from. The training dataset is drawn from the first 3 hospitals, while out-of-distribution validation and out-of-distribution test datasets are sampled from the 4th hospital and 5th hospital, respectively.

- **ISIC.** The melanoma dataset is from the International Skin Imaging Collaboration (ISIC) archive[7]. Data from the archive are collected by different organizations at different points in time [7–9, 17, 43, 45, 50]. There are about 70k data samples in total. In particular, the resized input image $x$ is a $224 \times 224$ image and a binary target label $y$ denotes whether the image exhibit is melanoma or not. The environment is the hospital from which the image was collected [8]. We follow a similar setup to Camelyon-17. The training dataset is drawn from the first 3 hospitals, while out-of-distribution validation and out-of-distribution test datasets are sampled from the 4th hospital and 5th hospital, respectively. For preprocessing, we filter out datapoints that are not specifically categorized as "benign" or "malignant" (e.g. "indeterminate"). The OOD validation dataset is from the "Barcelona1" site indicator and the OOD test dataset is the "Vienna1" site indicator.

- **FMoW.** The FMoW dataset is from the WILDS benchmark [6, 23], a satellite image classification task which includes 62 classes and 80 domains (16 years x 5 regions). Concretely, the input $x$ is a $224 \times 224$ RGB satellite image, the label $y$ is one of the 62 building or land use categories, and the environment represents the year that the image was taken as well

---

[7]https://www.isic-archive.com

[8]Hospitals are Hospital Clinic of Barcelona, Medical University of Vienna, University of Queensland Diamantina Institute, Memorial Sloan Kettering Cancer Center, University of Sydney Melanoma Diagnostic Centre, and University of Pittsburgh Medical Center.

as its corresponding geographical region – Africa, the Americas, Oceania, Asia, or Europe. The train/test/validation splits are based on the time when the images are taken. Specifically, images taken before 2013 are used as the training set. Images taken between 2013 and 2015 are used as the validation set. Images taken after 2015 are used for testing.

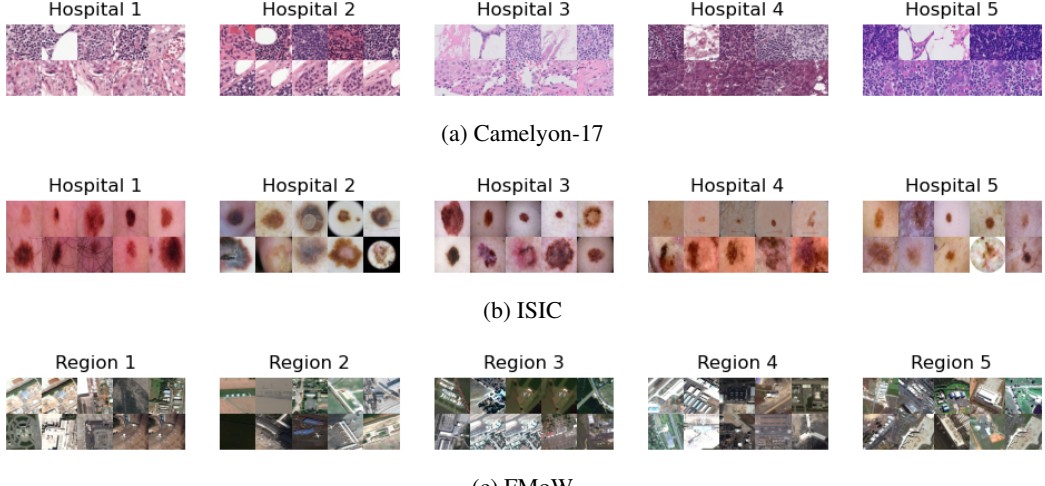

(a) Camelyon-17

(b) ISIC

(c) FMoW

Figure 4: Representative images for datasets, separated by domain. Each row depicts a separate class. For FMoW, for simplicity, we show 2 classes out of 62 and only images before 2013.

# C Hyperparameter Details

Table 3 shows hyperparameter settings for all datasets, where NW-specific hyperparameters are below the midline. For all models, we use pretrained ImageNet weights. For $\lambda$, we perform a grid-search over the values $\{0.01, 0.1, 1\}$. Fig. 5 depicts $\text{NW}_e^B$ performance vs $N_c$ for Camelyon-17 and ISIC datasets. We find that performance is relativity insensitive to $N_c$ above $\sim 5$ examples per class.

In the original NW head paper [55], the authors experiment with a temperature (i.e. bandwidth) hyperparameter $\tau$. In this work, we set $\tau = 1$ for all experiments. The reason for this is that we optimize both the feature extractor and classifier end-to-end, and the kernel used in the classifier is dependent on the features that the feature extractor learns (unlike [25], e.g.). Thus, we let the feature extractor to optimize the bandwidth on its own. Note this same approach is taken in prior works [47].

Table 3: Hyperparameter settings for various datasets.

| Hyperparameter | Camelyon-17 | ISIC | FMoW |
|---|---|---|---|
| Learning rate | 1e-4 | 5e-5 | 1e-4 |
| Weight decay | 1e-4 | 0 | 1e-2 |
| Scheduler | None | None | StepLR |
| Batch size | 32 | 8 | 8 |
| Architecture | DenseNet-121 | ResNet-50 | DenseNet-121 |
| Optimizer | SGD | Adam | Adam |
| Maximum Epoch | 10 | 5 | 60 |
| $N_q$ | 8 | 8 | 8 |
| $N_c$ | 8 | 8 | 1 |
| $N_s$ | $N_c \times 2 = 16$ | $N_c \times 2 = 16$ | $N_c \times 62 = 62$ |
| $\lambda$ | 0.01 | 0.01 | 0.1 |
| $k$ | 3 | 3 | 3 |

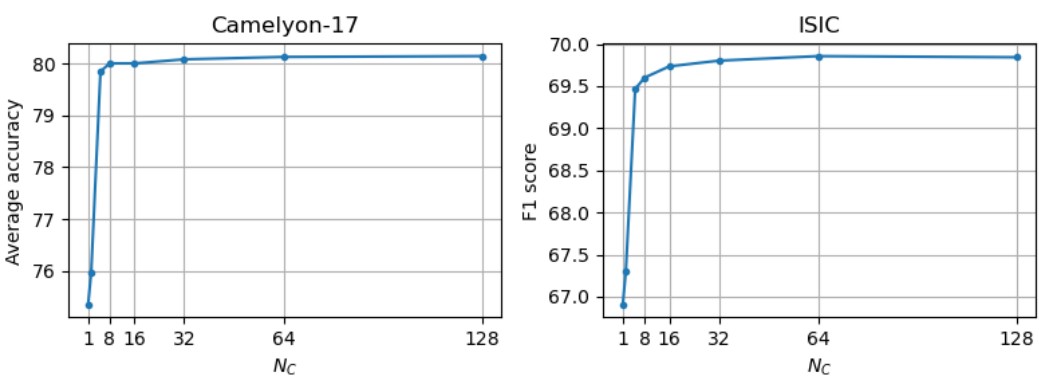

Figure 5: $\text{NW}_e^B$ performance vs $N_c$ for Camelyon-17 and ISIC datasets. Full mode. Performance is relativity insensitive to $N_c$ above $\sim 5$ examples per class.

# D Table of Runtimes

Table 4 shows approximate runtimes for various datasets during training and inference. All experiments are performed on a GPU.

Table 4: Approximate runtimes for various algorithms. Training time is time to complete maximum epochs as specified in Table 3, and does not include validation. Inference time is time to evaluate the entire test set. Averaged over all training runs.

|           | *Algorithm*   | *Camelyon-17* | *ISIC*  | *FMoW*  |
|-----------|---------------|---------------|---------|---------|
| Training  | ERM           | 7 hr          | 1 hr    | 22 hr   |
|           | NW            | 14 hr         | 2 hr    | 40 hr   |
| Inference | ERM           | 10 min        | 2 min   | 10 min  |
|           | NW, Random    | 15 min        | 3 min   | 20 min  |
|           | NW, Full      | 2 hr          | 15 min  | 1 hr    |
|           | NW, Ensemble  | 2 hr          | 15 min  | 1 hr    |
|           | NW, Cluster   | 2.2 hr        | 17 min  | 1.1 hr  |
|           | NW, Probe     | 10 min        | 2 min   | 10 min  |

# E ID vs. OOD Performance

With most invariant learning methods, prior works observe a tradeoff between in-distribution and out-of-distribution generalization. In Table 5, we show in-distribution (ID) and out-of-distribution (OOD) results for Camelyon-17, which provides an ID validation set. We observe that there is a tradeoff between ID and OOD performance, similar to prior work.

Table 5: In-distribution vs. out-of-distribution generalization performance on Camelyon-17.

|                  | *In-distribution* | *Out-of-distribution* |
|------------------|-------------------|-----------------------|
| ERM              | $93.2_{\pm 5.2}$  | $70.3_{\pm 6.4}$      |
| $NW^B$, Full     | $96.1_{\pm 1.0}$  | $72.0_{\pm 6.7}$      |
| $NW_e^B$, Full   | $92.8_{\pm 2.0}$  | $80.0_{\pm 2.7}$      |

# F  Imbalanced ISIC Experiments

As ISIC exhibits significant label imbalance where the positive class is much less represented than the negative class (see Fig. 6), we experiment with an NW variant without support-set label balancing as an ablation. For both variants, we set $N_c = 8$. To train the imbalanced variant, we sample a mini-batch support set by first sampling one image per class (to guarantee both classes are represented in the support at least once), and then sampling the rest of the images randomly from the dataset.

To characterize the performance of both variants across class imbalances, we change the prevalance of $y = 0$ in the test set by removing negative class images until the desired prevalence is achieved. Note the default prevalence is $\sim 0.85$. Then, we compute the accuracy over this manipulated test set (note, prior work has shown that F1-score is not a good metric for comparing classifiers with different label imbalances [3]).

Fig 7 shows results. We observe that at high prevalence where the proportion of negative classes matches in training and test, the imbalanced variant outperforms the balanced variant, whereas the opposite is true for low prevalence. This makes sense because at high prevalence, the class imbalance is similar for the training and test domains; thus, a model which overpredicts the negative class is usually right. On the other hand, this fails in test sets where the prevalence is flipped (i.e. low prevalence). These results suggest that $NW^B$ is a more robust classifier in the presence of label shift.

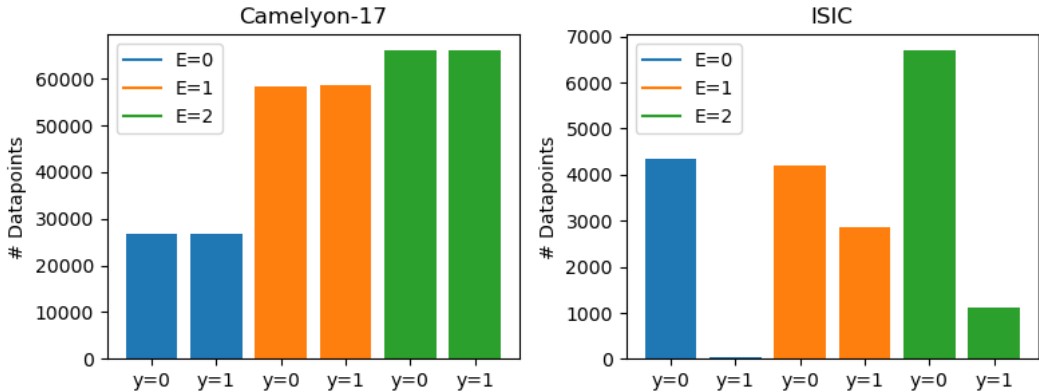

Figure 6: Number of datapoints separated by class for Camelyon-17 and ISIC datasets. There is significant label imbalance for the ISIC dataset.

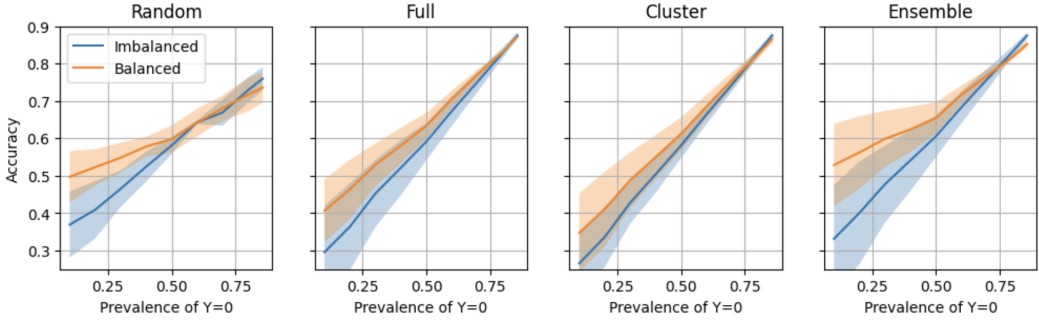

Figure 7: Accuracy of NW (imbalanced) and $NW^B$ (balanced) models over varying prevalence of $y = 0$ for ISIC dataset. At low prevalence where the prevalence differs the most from training domains, we observe that model performance is higher for $NW^B$. The default prevalence is 6705/7818 = 0.8576, which is the right-most value in the x-axis.

# G  Nearest-neighbor Inference Modes

In addition to Random, Full, Cluster, and Ensemble inference modes, we additionally experiment with two inference modes based on nearest neighbors: k-NN and HNSW. HNSW (Hierarchical Navigable Small Worlds) is a fast approximate nearest neighbor algorithm [33]. We choose $k = 20$ based on prior work [25]. HNSW is about two times faster in total runtime on a GPU than full k-NN.

Overall, we observe that k-NN and HNSW perform nearly identically for all the datasets and variants. Additionally, both modes perform better in terms of mean performance on Camelyon-17, and perform on par with the best-performing modes for ISIC (Cluster) and FMoW (Ensemble). However, they generally have higher variances across model runs. We suspect that less total samples used in the support (20 as compared to more than 1000 for Full) and the fact that not all classes are guaranteed to be represented in the support may lead to less stability across model runs.

Table 6: Metric average $\pm$ standard deviation for all datasets (%). Higher is better. Implicit / Explicit.

| Algorithm | Camelyon-17 | ISIC | FMoW |
|---|---|---|---|
| $NW^B$, Full | $72.0_{\pm 6.7}$ | $61.9_{\pm 3.5}$ | $31.6_{\pm 0.9}$ |
| $NW^B$, Cluster | $70.6_{\pm 6.9}$ | $61.4_{\pm 2.3}$ | $31.3_{\pm 0.9}$ |
| $NW^B$, Ensemble | $71.9_{\pm 6.0}$ | $63.9_{\pm 3.8}$ | $32.2_{\pm 1.0}$ |
| $NW^B$, k-NN | $72.5_{\pm 3.2}$ | $64.2_{\pm 2.6}$ | $32.5_{\pm 1.4}$ |
| $NW^B$, HNSW | $72.5_{\pm 3.2}$ | $64.2_{\pm 2.6}$ | $32.5_{\pm 1.4}$ |
| $NW_e^B$, Full | $80.0_{\pm 2.7}$ / $79.7_{\pm 1.9}$ | $69.6_{\pm 2.3}$ / $70.0_{\pm 1.0}$ | $35.0_{\pm 0.7}$ / $34.6_{\pm 0.4}$ |
| $NW_e^B$, Cluster | $78.6_{\pm 2.5}$ / $79.0_{\pm 1.4}$ | $71.1_{\pm 1.7}$ / $71.0_{\pm 1.0}$ | $33.9_{\pm 0.6}$ / $34.0_{\pm 0.3}$ |
| $NW_e^B$, Ensemble | $79.5_{\pm 2.6}$ / $79.6_{\pm 1.9}$ | $69.5_{\pm 2.2}$ / $69.8_{\pm 0.8}$ | $37.8_{\pm 0.9}$ / $38.2_{\pm 0.4}$ |
| $NW_e^B$, k-NN | $80.9_{\pm 10.3}$ / $80.2_{\pm 4.9}$ | $70.7_{\pm 7.4}$ / $71.0_{\pm 7.2}$ | $37.9_{\pm 2.3}$ / $38.3_{\pm 1.1}$ |
| $NW_e^B$, HNSW | $80.9_{\pm 10.3}$ / $80.2_{\pm 5.0}$ | $70.7_{\pm 7.4}$ / $71.0_{\pm 7.2}$ | $37.9_{\pm 2.2}$ / $38.2_{\pm 1.1}$ |

# H  Interpretability of NW Head

In this section, we provide both a visual and quantitative exploration of the interpretability capabilities of the NW head. In Fig. 8, we show several query images and its 8 nearest neighbors in the feature space, for both $NW^B$ and $NW_e^B$ variants. The colored border around each neighbor indicates which training environment the image comes from. Interestingly, we notice that the neighbors for $NW_e^B$ come from a variety of environments (note a variety of colored borders), while the neighbors for $NW^B$ are less diverse.

Fig. 9 quantifies this phenomenon, depicting a normalized histogram of the environments from which the top 20 nearest neighbors originate in the training dataset for Camelyon-17, averaged over all queries in the test set. From these results, it is clear that $NW_e^B$ leverages support images from a wider variety of environments to make its prediction, suggesting that it captures more invariant representations.

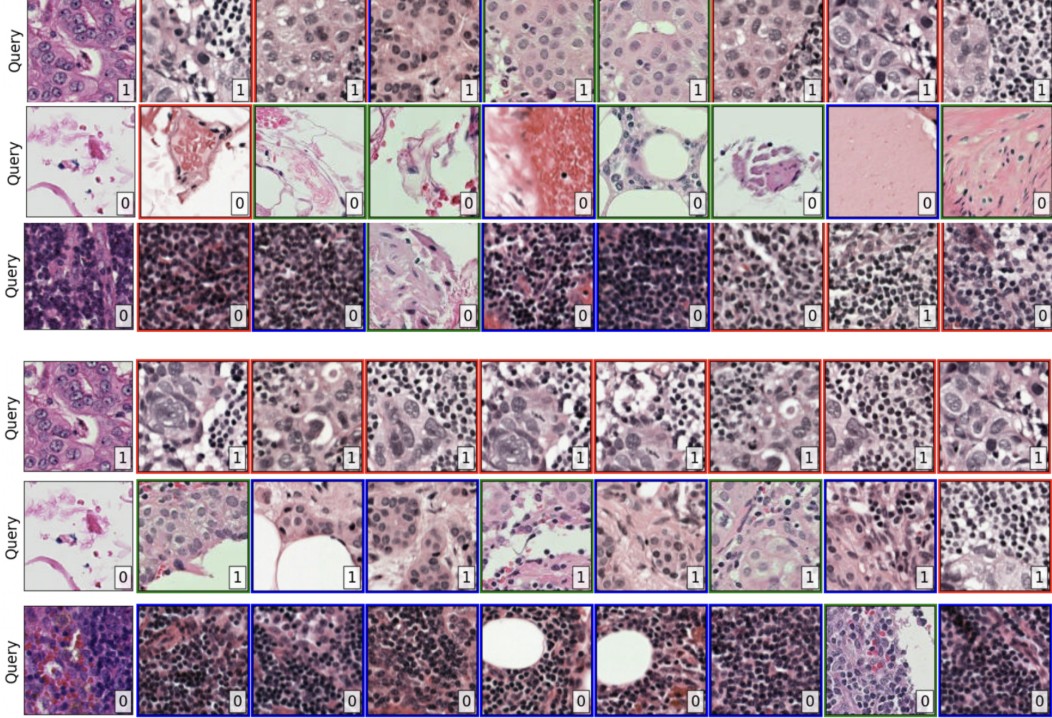

Figure 8: Visualization of 3 query images and their 8 nearest neighbors in the feature space, for both $NW_e^B$ (top) and $NW^B$ (bottom). Labels are shown in the bottom right corner, and colored borders of neighbors indicate which of the 3 training environments the image comes from. For both variants, we observe visual similarity between the query images and the nearest neighbors. However, we observe the neighbors for $NW^B$ tend to lack diversity in the environments from which they originate. In contrast, neighbors for $NW_e^B$ tend to come from a wider variety of environments, suggesting that $NW_e^B$ captures more invariant representations.

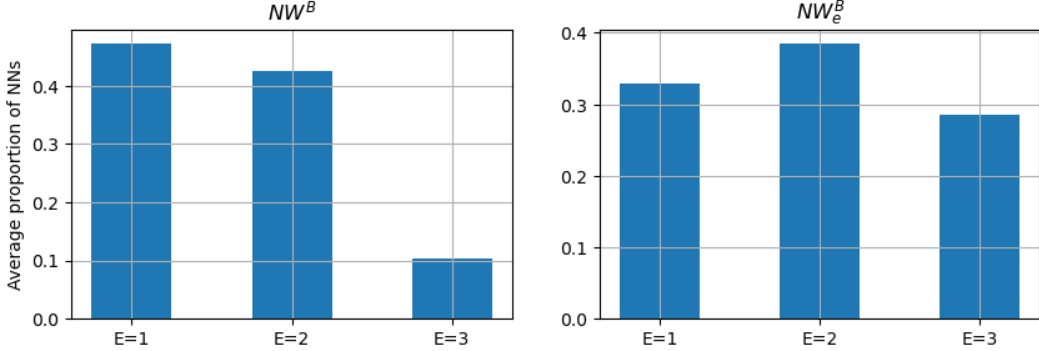

Figure 9: Normalized histogram of the environments from which the top 20 nearest neighbors originate in the training dataset for Camelyon-17, averaged over all queries in the test set. We observe a more balanced proportion for $NW_e^B$, indicating that the model relies more evenly across all 3 environments to make its prediction, and further suggesting that representations are more invariant than $NW^B$.

