# OpenReview forum: "Learning Invariant Representations with a Nonparametric Nadaraya-Watson Head"
_NeurIPS.cc/2023/Conference — NeurIPS 2023 poster_

### Official Review · Reviewer_f7MY · 2023-06-25

**Soundness:** 3 good
**Presentation:** 3 good
**Contribution:** 2 fair
**Rating:** 5
**Confidence:** 3

**Summary:**

This paper proposes a novel algorithm for learning invariant representations using a fixed head that is the sum of similarities of the query features with a set of support features, weighted by one-hot-encoded class label (in other words, the head predicts the class whose support features most align with the query features). Given this fixed head and a training dataset, the proposed method entails optimizing for a representation that solves a constrained maximum likelihood estimation problem. Experimental results evaluating the proposed technique on three benchmark datasets against a variety of baselines are provided.

**Strengths:**

1. The paper does a good job of formulating the problem and describing the proposed approach. The writing is generally clear.

2. The motivating application to medical image classification in different domains is strong, as this is an important and relevant unsolved problem.

3. The proposed approach is intriguing as it removes the bilevel complexity from the IRM problem, as it fixes a head rather than learning one.

4. The empirical results are rigorous and promising, as they show state-of-the-art or competitive performance by certain variations of the proposed method on three benchmark datasets.

5. The authors clearly acknowledge the limitations of the proposed method.

**Weaknesses:**

1. Practically speaking, the proposed method is computationally expensive because it requires computing pairwise comparisons between the representations support and query samples during training and inference, as the authors acknowledge. However, the support batch size is small during training, and the experiments show that using the Cluster method for inference can allow for doing inference with a small number of support samples without much drop in performance, meaning this is not a huge issue. Still, the empirical results are not overly impressive, as the performance of the best variant essentially matches the baseline [50]. It would be helpful to compare the computational costs of the proposed approach with [50], or run further experiments demonstrating a more clear advantage.

2. The motivating application is strong, and from this it is clear why we want estimators that satisfy conditions 1) and 2) in lines 127 and 128. It is also clear that the proposed estimator with the NW head satisfies these conditions. But it is not clear why we need the NW head in order to satisfy these; perhaps a simpler estimator that does not require using a support set of images for every query image could also satisfy 1) and 2).

3. The Introduction and/or Related Works would benefit from a more detailed discussion of the criticisms of IRM, especially relating to how the proposed method can alleviate these criticisms. Granted, IRM is compared with in Section 4.4, but this section does not discuss whether the proposed method addresses the concerns with IRM posed by [15,21,39].

---------

Post-rebuttal: I have raised my score, please see comment below.

**Questions:**

1. The statement "the objective is equivalent to unconstrained maximum likelihood under the assumption in Eq. 1" is not clear to me, since Eq. 1 only applies for some particular g_C.

2. The random inference mode does not make sense to me: how can the sampling be uniform across the dataset if each class is represented k times?

3. How are the features for NW Probe trained?

**Limitations:**

Yes.

---

> ### Author Rebuttal · Authors · 2023-08-08
>
> > Practically speaking, the proposed method is computationally expensive because… [...] It would be helpful to compare the computational costs of the proposed approach with [50], or run further experiments demonstrating a more clear advantage.
>
> We agree with the reviewer that computational cost is the primary limitation of our method. The computational costs of parametric approaches like [50] are very similar in practice to ERM, which we provide details of in Table 4.
>
> While we agree that empirical results of our proposed method are not substantially improved over [50], we believe our non-parametric regularization approach to enforcing invariance (whether explicit or implicit) is more natural and intuitive than baseline methods because an environment is encoded by manipulating the support set to contain real samples only from that environment. Other baseline methods use proxy methods to achieve this, for example by aligning calibration [2], intermediate layer activations [3], or variances [4] across environments.
>
> For other advantages of our method, we refer the reader to our general response.
>
> > The motivating application is strong, and from this it is clear why we want estimators that satisfy conditions 1) and 2) in lines 127 and 128. [...]. But it is not clear why we need the NW head in order to satisfy these; perhaps a simpler estimator…
>
> Satisfying 1) requires intervening on Y in the causal DAG. In practice, this can be achieved by matching the environment-specific prior for Y (e.g. by sampling balanced mini-batches during training). The approach we take is by balancing classes in the support set, which we view as a minor contribution that fits well with our theme of support set manipulation.
>
> Satisfying 2) essentially defines the invariant representation learning task and many prior works seek to satisfy it. Generally, this is achieved by regularizing the standard classification loss with constraints which align representations across environments (e.g. aligning correlations of layer activations or aligning model calibration). With the NW head, however, this can be achieved naturally, by manipulating the support set.
>
> > The Introduction and/or Related Works would benefit from a more detailed discussion of the criticisms of IRM, especially relating to how the proposed method can alleviate these criticisms. Granted, IRM is compared with in Section 4.4, but this section does not discuss whether the proposed method addresses the concerns with IRM posed by [15,21,39].
>
> Prompted by this suggestion, we have elaborated further in Section 4.4 discussing how the proposed method addresses the concerns with IRM. We provide an overview of these elaborations here.
>
> The IRM objective seeks a representation such that the optimal (parametric) linear classifier’s parameters on top of this representation is the same across all environments. The main concern raised by the aforementioned works (foremost in [21]) is the tractable version of this objective (IRMv1 in the IRM paper). This tractable objective assumes convexity and is complex, requiring computation of the Hessian. The NW head side-steps these issues by eliminating the need to learn the linear classifier parameters at all – instead, the classifier is *fixed by construction* since it has no learnable parameters. Essentially, we can enforce invariance through design of the support set, providing a much more intuitive and computationally simpler objective to optimize.
>
> Note that from an empirical standpoint, our experiments with NW-probe also address this issue. NW-probe can be directly compared against IRM, as both have exactly the same architecture and number of learnable parameters; additionally, the causal assumptions and the constraints imposed are theoretically identical. Interestingly, we find that NW-probe outperforms IRM in the datasets we try, suggesting that our $NW^B_e$ training strategy does learn better invariant representations.
>
> > The statement "the objective is equivalent to unconstrained maximum likelihood under the assumption in Eq. 1" is not clear to me, since Eq. 1 only applies for some particular g_C.
>
> Apologies for the confusion; our purpose in including this statement was to highlight the similarity between our objective and that of maximum likelihood estimation. Specifically, the particular $\phi$ and $g_C$ that satisfies the constraint in Eq. 5 (which is derived from Eq. 1) allows us to essentially drop the conditioning on $e_i$, which is the subscript for the probability estimator being maximized. Thus, we could equivalently write the objective as:
>
> $$
>    \arg\max_{\phi, g_C}  \sum_{i=1}^N \log \hat{P}^B (y_i \mid g_C(x_i); \phi)
> $$
> $$
> s.t. \
>     \hat{P}^B_e (y_i \mid g_C(x_i); \phi) = \hat{P}^B_{e'} (y_i \mid g_C(x_i); \phi), \ \ \forall i \in \{1, ..., N\}, \ \forall e, e'\in E.
> $$
>
>
> This formulation highlights the fact that our objective is simply maximum likelihood estimation with an additional invariance constraint.
>
> > The random inference mode does not make sense to me: how can the sampling be uniform across the dataset if each class is represented k times?
>
> Each class is represented k times, but which images from each class are present in the support set is chosen uniformly at random from the full training set.
>
> > How are the features for NW Probe trained?
>
> NW-probe is trained in two stages. In the first stage, we train the feature extractor weights ($\varphi$) only using an $NW^B_e$ training strategy (Eq. (7) or (8)). In the second stage, we freeze $\varphi$ and train a linear probe ($w$) on top of the frozen representations.

---

> ### Comment · Area_Chair_ECjF · 2023-08-16
> **Respond to authors' rebuttal**
>
> Please, look at the authors' rebuttal and the other reviewers' comments and indicate if you would like to change anything in your review.

---

> > ### Comment · Reviewer_f7MY · 2023-08-18
> >
> > Thank you to the authors for your detailed responses. After reading all the responses, I have a better understanding of the motivation for and demonstrated advantages of using the NW head. I have decided to raise my score from 4 to 5.

---

> > > ### Author Response · Authors · 2023-08-18
> > > **Response to reviewer f7MY**
> > >
> > > We thank reviewer f7MY for their prompt response and positive feedback.

---

### Official Review · Reviewer_xs7Q · 2023-07-06

**Soundness:** 2 fair
**Presentation:** 3 good
**Contribution:** 2 fair
**Rating:** 5
**Confidence:** 3

**Summary:**

The authors address an important problem of reliability in deep learning, given data collected from different sources (environments).
For this, the authors propose a method that allows the separation of style and content of input objects, which ensures stable behavior in different environments. The authors develop a causality graph and show how to embed causal assumptions into the model.
For a concrete model, the authors use a nonparametric method based on the Nadaraya-Watson (NW) head. For predictions, they leverage the NW head, which uses learned representations and a support set of labeled data.
They conduct experiments on several datasets with real-world environment variability, and experiments show the competitive performance of the approach.

**Strengths:**

1. The paper addresses an important problem and provides an interesting solution based on Nadaraya-Watson kernel regression, which is interesting.
2. In general, I find the paper well-written, well-structured, supplied with nice illustrations and easy to follow.
3. The choice of datasets for experiments is interesting and challenging.

**Weaknesses:**

1. The overall novelty and the extent of the contribution made by this paper are unclear. The use of the Nadaraya-Watson (NW) head on top of a feature extractor has been previously used [1, 2]. Furthermore, it's not clear whether the manipulation of the support set can be considered as a novel idea. It would be beneficial for the authors to explicitly state the specific contributions of their paper in the introduction section.
2. There appears to be a missing reference to Figure 3 in the paper.
3. No code available, so I can not check if experiments are reproducible.

[1] Alan Q. Wang and Mert R. Sabuncu. A flexible nadaraya-watson head can offer explainable and calibrated classification. Transactions on Machine Learning Research, 2023.

[2] Kotelevskii, Nikita, et al. "Nonparametric Uncertainty Quantification for Single Deterministic Neural Network." Advances in Neural Information Processing Systems 35 (2022): 36308-36323.

**Questions:**

1. It is well known in the literature related to the methods which work with embeddings (see, for example, [3, 4]) that there might be a feature collapse. So that for the small perturbation in the input, the change in the output is arbitrary. Did you use any techniques to prevent it, like spectral normalizations?
2. Typically, in kernel methods, there is a scale parameter (e.g., bandwidth) that is very important for performance and requires careful selection. As I see from Eq. 2, you simply use negative Euclidean distance. Is it safe? Might there be a case that embedding from one environment a scattered wider than from another, and it is required to use different scale parameters to fit them well?
3. In [2], authors also use NW head to a potentially big dataset, and to ease computations, they use approximate nearest neighbors algorithms. It would be interesting to apply this option here as well since the "random" inference scheme might be too noisy, "full" too expensive, and others too rough.


[3] Liu, Jeremiah, et al. "Simple and principled uncertainty estimation with deterministic deep learning via distance awareness." Advances in Neural Information Processing Systems 33 (2020): 7498-7512.

[4] van Amersfoort, Joost, et al. "On feature collapse and deep kernel learning for single forward pass uncertainty." arXiv preprint arXiv:2102.11409 (2021).




Edit: I would like to thank the authors for their answers during rebuttal period.

**Limitations:**

Authors discussed limitations in the main part of the paper.

---

> ### Author Rebuttal · Authors · 2023-08-08
>
> > The overall novelty and the extent of the contribution made by this paper are unclear…
>
> We have enumerated our specific contributions in the general response. To reiterate here, to the best of our knowledge, we believe that the manipulation of the support set for learning invariant features is a novel approach to invariant learning and the broader goal of domain generalization. We are not aware of any prior work that approaches invariant representation learning from the perspective of a manipulation of support set from a non-parametric perspective.
>
> In our revised manuscript, we plan to explicitly add a Contributions subsection in the Introduction section.
>
> > There appears to be a missing reference to Figure 3 in the paper.
>
> Thank you for pointing out this omission – we will add this missing reference in our revised manuscript.
>
> > No code available, so I can not check if experiments are reproducible.
>
> To maintain anonymity in the initial submission, we withheld a link to our code. We intend to make it available upon acceptance. Note that our code and dataset relies heavily on the NW head repository [1] and WILDS benchmark [6], thus facilitating easy reproduction and comparison. Both repositories are publicly available.
>
> > It is well known in the literature related to the methods which work with embeddings (see, for example, [3, 4]) that there might be a feature collapse. [...] Did you use any techniques to prevent it, like spectral normalizations?
>
> In our experiments, we have not observed the feature collapse phenomenon. Other works have found similar results; for example, the authors in [2] find little difference when adding spectral normalization (see Table 6 in [2]).
>
> The feature collapse phenomenon seems to arise in the context of Gaussian processes and its extension to deep networks, Deep Kernel Learning. These methods require complicated training schemes that minimize the ELBO, which can be decomposed into a data fit term and a complexity penalty term. From [4], the authors associate feature collapse with this complexity term. However, in the NW head, training is performed with a standard maximum-likelihood loss and thus is relatively simpler than GP-based methods. In particular, there is no complexity term which can cause feature collapse. Put another way, theoretically speaking, the NW head departs from standard parametric classification only from an architectural standpoint, and not from a training/inference/optimization standpoint.
>
> > Typically, in kernel methods, there is a scale parameter (e.g., bandwidth) that is very important for performance and requires careful selection….
>
> We did not tune the bandwidth in our experiments. The reason for this is that we optimize both the feature extractor and classifier end-to-end, and the kernel used in the classifier is dependent on the features that the feature extractor learns (unlike [2], e.g.). Effectively, we allow the feature extractor to optimize the bandwidth during training (note the same approach is taken in prior works, see [5]). Nevertheless, we take the reviewers point and believe the exploration of kernel bandwidths (and other kernels) might be an important future work (as we allude to in the Conclusion section Lines 296-297).
>
> We tend to disagree with the reviewer’s point that the embeddings from one environment might be scattered wider than another environment, since our optimization (Eqs. 7 or 8) enforces (either explicitly or implicitly) the embeddings from one environment to match the embeddings from another. Our additional results in Figure 9 shed some light on this hypothesis from an empirical perspective – we observe that the nearest neighbors of query images in the feature space come from all 3 training environments evenly. This indicates that the model relies more evenly across all 3 environments to make its prediction, and further suggests that representations are invariant across environments.
>
> > In [2], authors also use NW head to a potentially big dataset, and to ease computations, they use approximate nearest neighbors algorithms. It would be interesting to apply this option…
>
> Thank you for this interesting suggestion. Prompted by this, we have added these results in Table 5 in the corresponding PDF. In it, we show additional results for K-NN and HNSW (Hierarchical Navigable Small Worlds), a fast approximate nearest neighbor algorithm [7].
>
> Overall, we observe that k-NN and HNSW perform nearly identically for all the datasets and variants we tried. Additionally, both modes perform better in terms of mean performance on Camelyon-17, and otherwise perform about as well as the best-performing modes for ISIC (Cluster) and FMoW (Ensemble). However, they generally have higher variances across model runs. We suspect this is because there are less total samples used in the support (20 vs. more than 1000 for Full), and also not all classes are guaranteed to be represented in the support, leading to more unstable results.
>
>
> [1] Wang et al. A flexible nadaraya-watson head can offer explainable and calibrated classification, 2023.
> [2] Kotelevskii et al. Nonparametric Uncertainty Quantification for Single Deterministic Neural Network., 2022
> [3] Liu et al. Simple and principled uncertainty estimation with deterministic deep learning via distance awareness, 2020
> [4] van Amersfoort et al. On feature collapse and deep kernel learning for single forward pass uncertainty, 2021
> [5] Snell et al. Prototypical networks for few-shot learning, 2017
> [6] Koh et al. Wilds: A benchmark of in-the-wild distribution shifts, 2021.
> [7] Malkov et al. Efficient and robust approximate nearest neighbor search using Hierarchical Navigable Small World graphs, 2016.

---

> > ### Comment · Reviewer_xs7Q · 2023-08-16
> >
> > I would like to thank the authors for their detailed answers!
> >
> >
> > My questions and concerns are well addressed. I have no further questions for the authors, and I increase my evaluation score by one.

---

> > > ### Author Response · Authors · 2023-08-18
> > > **Response to reviewer xs7Q**
> > >
> > > We thank reviewer xs7Q for their prompt response and positive feedback.

---

> ### Comment · Area_Chair_ECjF · 2023-08-16
> **Respond to authors' rebuttal**
>
> Please, look at the authors' rebuttal and the other reviewers' comments and indicate if you would like to change anything in your review.

---

### Official Review · Reviewer_48ug · 2023-07-06

**Soundness:** 3 good
**Presentation:** 3 good
**Contribution:** 2 fair
**Rating:** 5
**Confidence:** 4

**Summary:**

The authors apply a recently proposed method for similarity-based prediction to the problem of invariant learning. This paper builds off of the Nadaraya-Watson architecture where predictions for a test input are derived via nearest proximity according to a learned kernel. The proposed method uses a NW head for each training environment, plus some additional regularization to encourage invariance over an assumed causal graph.


**Strengths:**

Regularization-based approaches to invariant learning are known to entail optimization difficulties [https://proceedings.mlr.press/v162/zhang22u.html]. The non-parametric prediction approach taken by the authors provides a unique and interesting alternative.

The proposed method shows promise on relevant datasets. I was especially interested in the probe variant of NW-training, which indicates a more invariant internal representation than IRM.


**Weaknesses:**

While the proposed approach is novel and interesting, after reading the paper I was still left with questions as to how the method is implemented, and why it works (see questions below). Also, I think the paper could benefit from some simple theoretical analysis that demonstrates (even in a simplified setting) when we expect NW-based training will discover invariant features. I also feel that the proposed approach, which uses causal graphs to motivate independances on subsets of the training data, is very related to recently proposed MMD-based approaches [https://proceedings.mlr.press/v151/makar22a, https://arxiv.org/abs/2209.09423]. A discussion on how the proposed method (and its use of non-parameteric prediction) differs from these papers would be useful.

The fact that so many different variants of the NW method are tried makes me a bit wary of claims like "NW^B does X percent better than ERM" [line 256]. Not all the NW^B flavors outperform ERM.


**Questions:**

* What is the role of the support set in OOD generalization? Are training samples always used to make predictions at test time? Empirical evidence suggests OOD generalization is possible, but since the method relies on domain-labeled training examples to make predictions, is there some implicit assumption about how domains are related? For example in the IRM paper they discuss the relationship between domains geometrically in terms of “linear general position”.
* How is the kernel bandwidth (called "temperature" in the NW head paper) chosen?  This seems like a critical hyperparameter but I didn't see any discussion of it.
* Why is it more effective to regularize a non-parametric predictor (Eqn 7) than to directly regularize a standard classification loss?
* With most invariant learning methods there is a tradeoff between in-distribution and out-of-distribution generalization. Do you see that pattern with your method? Table 2 only reports OOD generalization if I understand correctly

Misc comments:
* The proposed method “leverages the NW head as a conditional estimator for Y conditioned on Z” [143–144]. The workshop paper “Towards Environment-Invariant Transfer..." by Eyre et al (https://openreview.net/forum?id=c4l4HoM2AFf) may be of interest because they also use kernels to estimate similar statistics of the learned representations.


**Limitations:**

Limitations are addressed [Sec 6].

---

> ### Author Rebuttal · Authors · 2023-08-08
>
> > [...]. A discussion on how the proposed method differs from these papers would be useful.
>
> We agree with the reviewer and reiterate that our causal setup and DAG in Fig. 2b is not novel, and has been proposed in many works within this literature (see also [1-2]). As mentioned in the general response, our primary contribution in this work is manipulation of the support set in an NW head during training for learning invariant and robust representations, and for the broader purpose of domain generalization. To the best of our knowledge, ours is the first work to approach domain generalization in a non-parametric manner, as well as the first work to experiment with manipulating the support set to inject prior information into the training process.
>
> > The fact that so many different variants of the NW method are tried makes me a bit wary of claims like "NW^B does X percent better than ERM" [line 256]. Not all the NW^B flavors outperform ERM.
>
> To clarify, we only make the claim that NW^B outperforms ERM on the imbalanced ISIC dataset, where balancing theoretically offers a potential OOD advantage. For other more balanced datasets (Camelyon-17 and FMoW), NW^B does not improve over ERM, as one might expect.
>
> To demonstrate this further, we have trained an ERM variant with balanced classes per environment for ISIC, which we denote $ERM^B$, and present the results below.
>
> |      | F1 score |
> | ----------- | ----------- |
> | $ERM$ | 58.2 (2.9)|
> | $NW^B$, Ensemble|63.9 (3.8)|
> | $ERM^B$|63.0 (2.5)|
>
> We find that performance is on-par with $NW^B$, as is expected.
>
> > What is the role of the support set in OOD generalization? Are training samples always used to make predictions at test time?...is there some implicit assumption about how domains are related?
>
> The reviewer is correct in that the support set is composed of the training set in order to make predictions at test time. Note that we only rely on domain-labeled training examples to make predictions in Ensemble inference mode, and not any other inference mode (Random, Full, and Cluster). For Camelyon and ISIC datasets, the use of domain information at prediction doesn’t make much difference in optimal performance (e.g. compare $NW^B_e$ performance for both Full and Ensemble).
>
> Theoretically, we do not make any assumptions about how domains are related other than assuming that if any X has a non-zero probability in one environment, it has a non-zero probability in all environments (this is akin to the positivity assumption in causal inference). We agree with the reviewer that this is an important question that, while we believe is out of scope for our current manuscript, should and has been further explored in other works (e.g. see [1])
>
> > How is the kernel bandwidth (called "temperature" in the NW head paper) chosen? This seems like a critical hyperparameter but I didn't see any discussion of it.
>
> The bandwidth hyperparameter (i.e. temperature) was set to 1 for all experiments. Thank you for pointing out this omission – we have added this detail in the revised manuscript.
>
> We did not tune this hyperparameter. The reason for this is that we optimize both the feature extractor and classifier end-to-end, and the kernel used in the classifier is dependent on the features that the feature extractor learns (unlike [6], e.g.). Thus, we let the feature extractor to figure out the bandwidth on its own (note the same approach is taken in prior works, see [5]). Nevertheless, we take the reviewers point and believe the exploration of kernel bandwidths (and other kernels) and its relation to optimization procedures might be an important future work (as we allude to in the Conclusion section, lines 296-297).
>
> > Why is it more effective to regularize a non-parametric predictor (Eqn 7) than to directly regularize a standard classification loss?
>
> We note that the first term in Eq. 7 is a standard cross-entropy classification loss. The second term in Eq. 7 enforces invariance across environments by forcing predictions from two different environments to be the same for a given input. We argue that our non-parametric regularization approach to enforcing invariance is more natural and intuitive than baseline methods because an environment is encoded by manipulating the support set to contain real samples only from that environment. Other baseline methods use proxy methods to achieve this, for example by aligning calibration [2], intermediate layer activations [3], or variances [4] across environments.
>
> However, as we mention in our general response, the result we find the most compelling in this work is the Implicit variant (NWbe-Implicit), which is competitive with and often outperforms state-of-the-art baselines while requiring no hyperparameter to tune. Such an implicit approach is difficult to achieve with a parametric strategy.
>
> > With most invariant learning methods there is a tradeoff between in-distribution and out-of-distribution generalization. Do you see that pattern with your method?
>
> Below, we show in-distribution (ID) and out-of-distribution (OOD) results for Camelyon-17, which provides an ID validation set.
>
>
> |      | ID | OOD |
> | ----------- | ----------- | ----------- |
> | $ERM$|93.2 (5.2)|70.3 (6.4)|
> | $NW^B$, Full|96.1 (1.0) |72.0 (6.7)|
> | $NW^B_e$, Full|92.8 (2.0)| 80.0 (2.7)|
>
> We observe that there is a tradeoff between ID and OOD performance, similar to prior work. We plan to add these results to our revised manuscript.
>
> [1] Rosenfeld et al. The risks of invariant risk minimization, 2021.
> [2] Wald et al. On calibration and out-of-domain generalization, 2021
> [3] Sun et al. Deep coral: Correlation alignment for deep domain adaptation, 2016
> [4] Krueger et al. Out-of-distribution generalization via risk extrapolation (rex), 2021.
> [5] Snell et al. Prototypical networks for few-shot learning, 2017.
> [6] Kotelevskii et al. Nonparametric Uncertainty Quantification for Single Deterministic Neural Network. 2022.

---

> > ### Comment · Reviewer_48ug · 2023-08-10
> > **acknowledging author rebuttal**
> >
> > I have read the author response and other reviews. Thank you to the authors for acknowledging my concerns and helping me to better understand the implementation (temperature selection, etc.).
> >
> > I don't think we are on the same page about the ISIC results. Even if I look at just that column, I see some $NB^B$ flavors that do better than ERM, and some that do worse. I really appreciate the completeness of the experiment design w.r.t. trying different baselines and variants of the proposed method. I'm just not totally comfortable with how the results are described. Correct me if I'm missing something.
> >
> > I think the paper will indeed be stronger by discussing more related works. But the submission and rebuttal as-is, especially given that a theoretical analysis of when NW can succeed is out of scope, is not enough to move the needle for me. I maintain my score.

---

> > > ### Author Response · Authors · 2023-08-10
> > > **Clarification about $NW^B$ results**
> > >
> > > We thank the reviewer for their prompt response.
> > >
> > > Apologies for the confusion about the $NW^B$ results. Looking at the table of results for the ISIC dataset, the reviewer is correct that Random mode for $NW^B$ (56.7) is indeed lower than ERM (58.2).
> > >
> > > Random mode is the worst-performing of all the inference modes in general. This might be expected as it uses the least number of elements per support and thus has the least support information to make a prediction. We agree that this should be specified clearly in the manuscript, which we will update. Sorry again for the confusion!

---

### Official Review · Reviewer_d2Ui · 2023-07-06

**Soundness:** 2 fair
**Presentation:** 4 excellent
**Contribution:** 2 fair
**Rating:** 5
**Confidence:** 3

**Summary:**

This works proposes a causally motivated new setting for domain generalization building on the existing nonparametric Nadaraya-Watson head on top of a learned neural encoder. More precisely, it assumes that data inputs are causally generated from style latent features, which are environment-dependent, and content latent features, which are environment-independent after class-wise balancing. The authors hence try to learn these content features with a neural network. Predictions are made using the classical Nadaraya-Watson classifier on the learned representations of some support points. Training is carried with gradient-based maximum-likelihood, both with and without an additional penalty that encourages environment-independence of representations. At inference, fixed support points from the training set are chosen following one out of four strategies. The proposed method is compared to well-known domain generalization approaches in three datasets.

**Strengths:**

### Originality

The Nadaraya-Watson head is not novel, nor the causal framework proposed for domain generalization. Hence, the originality of the paper lies in using this nonparametric method for this purpose, motivated by the fact that the choice of the support set provides “a degree of flexibility not possible with parametric models”. This seems enough to say that the work is original and potentially significant for the subfield.

### Clarity

I found the paper very well-written and structured. Figures are clean and the method description is clear.

### Quality

The paper presents interesting results in a rather fair setting, with error bars and well explained experimental protocol.

### Significance

This work addresses the very important and difficult question of domain generalization with a very creative approach: non-parametric learning coupled with neural representations.

**Weaknesses:**

### Clarity

The only remark I would have is that the difference between NWb and NWBe-implicit could be better explained and highlighted in the paper (it took me a while to understand the subtle difference between them).

Typos:

- l. 255 “and”


### Quality

Despite the very positive points listed above, I still have a few concerns (from the more important to the least):

1. My main remark here concerns the pros and cons of the method, which are not very well highlighted in my opinion. While the idea is very interesting and it is easy to understand why having access to support points can help for domain generalization, I don’t think this is very well illustrated in the experiments. Results from table 2 show that NWbe is equivalent to CLOvE in Camelyon-17, worse than it in FMoW and (given the high uncertainties) only slightly better than it on ISIC (the only dataset on which CLOvE was reimplemented, according to the Appendix). Given the important computational caveats of the NWbe method (2x longer training and 8-16x longer inference than ERM) and additional overhead (double or triple batches, additional hyperparameters like number of cluster, strategy, etc), I find important to highlight and illustrate what benefits the method has compared to other existing methods which lead to comparable results. For example, it is mentioned in the conclusion that the method has interpretability advantages but this is not shown in the experiments.
2. Another point that surprised me was the fact that the implicit training (without regularization) led to comparable or better results than the regularized version. How do you explain that? I believe this point needs more investigation as it could indicate that the reason your method works is not exactly the ones explained (learning of environment-independent content latent features). It could be interesting for example to evaluate for both implicit and explicit models the invariance of learned neural embeddings to environment changes.
3. Is it fair to compare IRM to your representations + MLP given that IRM is trained with a simple linear classifier?
4. In figure 7 of the appendix, the rightmost accuracies for NWb (close to 90%) are substantially higher than those reported in table 2 (around 60%). What is the reason?
5. Also regarding this figure 7, I am curious why the balanced plot increases quite fast as the test set becomes more and more imbalanced. It is a bit surprising given that the method consists in class-balancing at training isn’t it ? Have you investigated why?
6. Also regarding balancing, I think that an additional baseline corresponding to ERM with class-balancing could be useful to better understand what is bringing the deltas in performance between ERM and NWb in table 2: is it the balancing or the NW head or both?
7. How was the number of clusters $k$ set? I see in appendix it was set to 3, but an analysis of its impact or at least a comment explaining why such hyperparameter is not a sensitive one could be a plus.

### Significance

Despite the very interesting method and important questions addressed, I think the paper experimental part still needs a little work.

**Questions:**

Question above

**Limitations:**

Limitations of the method are cited and discussed in the appendix.

---

> ### Author Rebuttal · Authors · 2023-08-08
>
> > The only remark I would have is that the difference between NWb and NWBe-implicit could be better explained and highlighted in the paper.
>
> We plan to add additional clarifying statements in our revised manuscript. To be clear, the difference between them is that in $NW^B_e$-implicit, we not only balance across classes (B), but also condition the support on an environment (e). The (-implicit) indicates that we don’t explicitly regularize the domain invariance (Eq. 7), but enforce it implicitly through the training process (Eq. 8).
>
> > My main remark here concerns the pros and cons of the method, which are not very well highlighted in my opinion. [...]
>
> Thank you for raising this important question. We have addressed it in the general response, and kindly refer the reviewer there.
>
> > Another point that surprised me was the fact that the implicit training (without regularization) led to comparable or better results than the regularized version. How do you explain that?
>
> In Implicit (Eq. 8), the constraint will be approximately satisfied in the sense that the model will be encouraged to predict the identical ground truth across all environments for an input. As an example, consider a classification task with 3 environments, and consider 3 images from the 3 environments each with label 0. Given infinite data and infinite model capacity, the model will predict a label 0 for each of these images, thus achieving invariance across the 3 domains.
>
> This analysis is borne out in the experiments. For the simpler datasets (Camelyon-17 and ISIC), Implicit performs equally well with higher variances compared to Explicit. In contrast, for FMoW, Explicit outperforms Implicit on Ensemble mode. This could be because the model does not have enough capacity to capture the invariance in an implicit setting.
>
> > Is it fair to compare IRM to your representations + MLP given that IRM is trained with a simple linear classifier?
>
> IRM and NW-probe ($NW^B_e$-learned representations + finetuned MLP) have exactly the same architecture. In IRM, both the feature extractor weights ($\varphi$) and the classifier weights ($w$) are trained simultaneously. In contrast, NW-probe is trained in two stages. In the first stage, we train the feature extractor weights ($\varphi$) only using an $NW^B_e$ training strategy (Eq. (7) or (8)). In the second stage, we freeze $\varphi$ and train a linear probe ($w$) on top of the frozen representations.
>
> NW-probe provides value as an academic exercise, as it can be directly compared with IRM; the causal assumptions and constraints are theoretically identical. Interestingly, we find that it outperforms IRM in the tasks we try, suggesting that $NW^B_e$ training strategy learns better invariant representations.
>
> > In figure 7 of the appendix, the rightmost accuracies for NWb (close to 90%) are substantially higher than those reported in table 2 (around 60%). What is the reason?
>
> The reason for this difference is that we report F1 scores in Table 2, but we report accuracies in Figure 7. As we mention in Lines 525-6, [2] finds that F1 score is not a good metric for comparing models with different label imbalances. In practice, we find graphs akin to Figure 7 but using F1 scores are not as striking as using accuracies.
>
> > Also regarding this figure 7, I am curious why the balanced plot increases quite fast as the test set becomes more and more imbalanced. It is a bit surprising given that the method consists in class-balancing at training isn’t it ? Have you investigated why?
>
> Note that in Fig. 7, a prevalence value of 0.25 indicates that 25% of samples in training are Y=0. Thus, the plot sweeps from low prevalence values for Y=0 (~25%) to high prevalence values (>75%). Note that the test-set prevalence is ~85%, so the right-most side of Fig. 7 shows performance when the train and test set prevalences are close.
>
> We agree with the reviewer that the Balanced (orange) plot increases as the prevalence of label Y=0 increases, although it doesn’t increase as quickly as Imbalanced (blue). Thus, while $NW^B$ mitigates label shift to an extent, increasing the prevalence of Y=0, thereby matching the train/test prevalences, leads to improved performance. However, we observe that the highest gap between the orange and blue lines are at low prevalence values. Thus, when prevalence values between train and test are maximally different, $NW^B$ proves to be a good robust predictor.
>
>
> > Also regarding balancing, I think that an additional baseline corresponding to ERM with class-balancing could be useful…
>
> In response to this comment, we have trained an ERM variant with balanced classes per environment for ISIC, which we denote $ERM^B$, and present the results below.
>
> |      | F1 score |
> | ----------- | ----------- |
> | $ERM$ | 58.2 (2.9)|
> | $NW^B$, Ensemble|63.9 (3.8)|
> | $ERM^B$|63.0 (2.5)|
>
> We find that performance is on-par with $NW^B$. This is expected as the theoretical assumptions are the same for both models; that is, removing dependence of E on Y via class balancing. We plan to add these results to our revised manuscript.
>
> > How was the number of clusters set?
>
> A value of $k=3$ was chosen as it seemed to balance good error rate performance (see row 1 of Figure 2 in [1]) with computational efficiency. In [1], accuracy is shown to be stable for varying values of $k$. We plan to add this detail to our revised manuscript.
>
> > Despite the very interesting method and important questions addressed, I think the paper experimental part still needs a little work.
>
> We hope that our additional results and explanations regarding experimental details provide a clearer picture of our work.
>
> [1] Alan Q. Wang and Mert R. Sabuncu. A flexible nadaraya-watson head can offer explainable and calibrated classification. Transactions on Machine Learning Research, 2023.
> [2] Jan Brabec, Tomáš Komárek, Vojtech Franc, and Lukáš Machlica. On model evaluation under non-constant class imbalance, 2020.

---

> > ### Comment · Reviewer_d2Ui · 2023-08-12
> > **Answer to authors**
> >
> > I would like to thank the authors for their detailed and thoughtful rebuttal. My concerns were well addressed as long as the proposed clarifications are added to the manuscript. Just to make sure my comment concerning IRM is clear, I think that saying you “finetune a” and **linear classifier** instead of a "**fully connected classifier**“ in l.205 would be clearer and avoid doubts concerning fairness.

---

> > > ### Author Response · Authors · 2023-08-18
> > > **Response to reviewer d2Ui**
> > >
> > > We thank the reviewer for their prompt response and positive feedback. We will be sure to change the wording in l.205 as suggested by the reviewer, as we agree this is clearer.

---

> > > > ### Comment · Reviewer_d2Ui · 2023-08-19
> > > > **Rating**
> > > >
> > > > As my peers, I am raising my rating from 4 to 5. Thanks again to the authors for their clarifications.

---

### Author Rebuttal · Authors · 2023-08-08

We thank the reviewers for their positive and constructive feedback. Here, we provide a general response to all reviewers, and provide a point-by-point response to each reviewer’s comment/questions below their corresponding review.

### Contributions

As raised by reviewers 48ug and ​​xs7Q, we enumerate the contributions of our paper:
1. Our primary contribution is manipulation of the support set in an NW head during training for learning invariant representations, and for the broader purpose of domain generalization. To the best of our knowledge, ours is the first work to approach domain generalization in a non-parametric manner, as well as the first work to experiment with manipulating the support set to inject prior information into the training process.
2. Another (minor) contribution is that we motivate label shift and IRM simultaneously through d-separation in a causal DAG. To the best of our knowledge, ours is the first to model label shift as an intervention on the Y node in the DAG in Fig. 2a. In our experiments, we find that both lead to a performance boost. Our non-parametric method enforces both these constraints in one model.

We clarify that the NW head itself is not a novel contribution, nor is our causal setup related to IRM (Fig. 2b), which has been proposed in many prior works.

### Advantages of the NW Head

As raised by reviewers d2Ui and f7MY, we enumerate the advantages of NW over baselines and refer the reader to Section 6 of our paper for a discussion of its disadvantages:
1. The Implicit variant (NWbe-Implicit) has no hyperparameter to tune. This variant is competitive with and often outperforms state-of-the-art baselines which all require tuning a hyperparameter coefficient in the regularized loss.
2. The NW head enables interpretability by interrogating nearest neighbors in the feature space. Since these neighbors directly contribute to the model’s prediction (Eq. 2), interrogation enables a user to see what is driving the model’s decision-making. This not only allows for greater model transparency, but also enables interrogating the quality of the invariant features. See Figs. 8 and 9 in the included PDF and “Additional Changes” below. Note this is not possible with parametric baselines, see [1].
3. Intuitively, we believe our non-parametric approach to enforcing invariance across environments is more natural and intuitive than baseline methods because an environment is encoded by manipulating the support set to contain *real samples* only from that environment. Other baseline methods resort to proxy methods to enforce invariance [3-5].

### Additional Changes

1. Prompted by reviewer d2Ui, in Figs. 8 and 9 in the included PDF, we provide both a visual and quantitative exploration of the interpretability capabilities of the NW head. In Fig. 8, we show several query images and its 8 nearest neighbors in the feature space, for both $NW^B$ and $NW^B_e$ variants. We add a colored border around each neighbor to indicates which training environment it comes from. Interestingly, we notice that the neighbors for $NW^B_e$ come from a variety of environments (note a variety of colored borders), while the neighbors for $NW^B$ are less diverse. Fig. 9 quantifies this phenomenon. Thus, $NW^B_e$ leverages support images from a wider variety of environments to make its prediction, suggesting that it captures more invariant representations.

2. Prompted by reviewer xs7Q, we have experimented with 2 additional inference modes: k-NN and HNSW (Hierarchical Navigable Small Worlds), a fast approximate nearest neighbor algorithm [6]. These results are shown in Table 5 in the included PDF. We choose $k=20$ based on prior work [2]. HNSW is about 2x faster in total runtime on a GPU than full k-NN; we plan to add these findings to our computational results in Table 4. Overall, we observe that k-NN and HNSW perform nearly identically for all the datasets and variants. Additionally, both modes perform better in terms of mean performance on Camelyon-17, and perform on par with the best-performing modes for ISIC (Cluster) and FMoW (Ensemble). However, they generally have higher variances across model runs. We suspect that less total samples used in the support (20 vs. more than 1000 for Full) and the fact that not all classes are guaranteed to be represented in the support may lead to more unstable results.

### Further Directions

Finally, we would like to highlight the future directions that our work may inspire. In general, training-time manipulation of a nonparametric support provides a novel means of imposing prior knowledge. Intuitively, every manipulation *creates a new classification problem*. In this work, we explored 2 such sources of prior knowledge: environment/domain knowledge, and assumption of balanced labels. We show that encoding these priors in a non-parametric way results in comparable to superior results compared to methods which encode priors via a regularized loss, and in certain cases obviates the need for tuning the associated hyperparameter.

Further research may explore other ways to leverage this new degree of flexibility. As we mention in the Conclusion, an interesting way to exploit any prior knowledge of class imbalances might be to upweight the occurrence of samples for more prevalent labels during training.

[1] Wang et al. A flexible nadaraya-watson head can offer explainable and calibrated classification. Transactions on Machine Learning Research, 2023.
[2] Kotelevskii et al. Nonparametric Uncertainty Quantification for Single Deterministic Neural Network, 2022.
[3] Wald et al. On calibration and out-of-domain generalization, 2021
[4] Sun et al. Deep coral: Correlation alignment for deep domain adaptation, 2016
[5] Krueger et al. Out-of-distribution generalization via risk extrapolation (rex), 2021.
[6] Malkov et al. Efficient and robust approximate nearest neighbor search using Hierarchical Navigable Small World graphs, 2016.

---

### Decision · Program_Chairs · 2023-09-21

**Decision:**

Accept (poster)

**Comment:**

Summary:

The authors address an important problem of reliability in deep learning, given data collected from different sources (environments). For this, the authors propose a method that allows the separation of style and content of input objects, which ensures stable behavior in different environments. The authors develop a causality graph and show how to embed causal assumptions into the model. For a concrete model, the authors use a nonparametric method based on the Nadaraya-Watson (NW) head. For predictions, they leverage the NW head, which uses learned representations and a support set of labeled data. They conduct experiments on several datasets with real-world environment variability, and experiments show the competitive performance of the approach. Experimental results evaluating the proposed technique on three benchmark datasets against a variety of baselines are provided.

Strengths:

1 - The work is original and potentially significant for the subfield.

2 - The paper is very well-written and structured. Figures are clean and the method description is clear.

3 - The paper presents interesting results in a rather fair setting, with error bars and well explained experimental protocol.

4 - This work addresses the very important and difficult question of domain generalization with a very creative approach: non-parametric learning coupled with neural representations.

5 - Regularization-based approaches to invariant learning are known to entail optimization difficulties. The non-parametric prediction approach taken by the authors provides a unique and interesting alternative.

6 - The proposed method shows promise on relevant datasets. I was especially interested in the probe variant of NW-training, which indicates a more invariant internal representation than IRM.

7 - The paper addresses an important problem and provides an interesting solution based on Nadaraya-Watson kernel regression, which is interesting.

8 - The choice of datasets for experiments is interesting and challenging.

9 - The paper does a good job of formulating the problem and describing the proposed approach. The writing is generally clear.

10 - The motivating application to medical image classification in different domains is strong, as this is an important and relevant unsolved problem.

11 - The proposed approach is intriguing as it removes the bilevel complexity from the IRM problem, as it fixes a head rather than learning one.

12 - The empirical results are rigorous and promising, as they show state-of-the-art or competitive performance by certain variations of the proposed method on three benchmark datasets.

13 - The authors clearly acknowledge the limitations of the proposed method.

Weaknesses:

1 - The pros and cons of the method are not very well highlighted in my opinion.

2 - The paper experimental part still needs a little work.

3 - After reading the paper one is still left with questions as to how the method is implemented, and why it works.

4 - The fact that so many different variants of the NW method are tried makes one a bit wary of claims like "NW^B does X percent better than ERM" .

5 - The overall novelty and the extent of the contribution made by this paper are unclear.

6 - No code available, so I can not check if experiments are reproducible.

7 - The proposed method is computationally expensive because it requires computing pairwise comparisons between representations support and query samples.

8 - The empirical results are not overly impressive.

9 - It is not clear why we need the NW head.

10 - The Introduction and/or Related Works would benefit from a more detailed discussion of the criticisms of IRM.

Decision:

All reviewers vote for acceptance. I, therefore, decide to accept the paper and encourage the authors to use the feedback provided to improve the paper for its camera ready version.